# Potential Roles of Exosomes in the Development and Detection of Malignant Mesothelioma: An Update

**DOI:** 10.3390/ijms232315438

**Published:** 2022-12-06

**Authors:** Phillip Munson, Arti Shukla

**Affiliations:** Department of Pathology and Laboratory Medicine, College of Medicine, University of Vermont, Burlington, VT 05405, USA

**Keywords:** exosomes, malignant mesothelioma, asbestos

## Abstract

Malignant mesothelioma (MM) is a devastating cancer of mesothelial cells, caused by asbestos exposure. Limited knowledge regarding the detection of asbestos exposure and the early diagnosis of MM, as well as a lack of successful treatment options for this deadly cancer, project an immediate need to understand the mechanism(s) of MM development. With the recent discovery of nano-vesicles, namely exosomes, and their enormous potential to contain signature molecules representative of different diseases, as well as to communicate with distant targets, we were encouraged to explore their role(s) in MM biology. In this review, we summarize what we know so far about exosomes and MM based on our own studies and on published literature from other groups in the field. We expect that the information contained in this review will help advance the field of MM forward by revealing the mechanisms of MM development and survival. Based on this knowledge, future therapeutic strategies for MM can potentially be developed. We also hope that the outcome of our studies presented here may help in the detection of MM.

## 1. Introduction

Few areas of research have grown as quickly and with as much enthusiasm as that of extracellular vesicle research related to exosomes. Exosomes are small, 40–140 nm membrane bound vesicles secreted from cells and originating from the endosomal pathway. These vesicles are enriched in biologically functional molecules (proteins, mRNA, miRNA, DNA, and lipids) and are vital to intercellular communication [1]. The communication conduit established by exosome transport from producer cells to target cells is important to normal physiology as well as disease states such as cancer [2].

As exosomes exist as a part of complex biosystem, their isolation, detection, and characterization is a challenging process. A recent review by Xu et al. (2022) provides an excellent resource related to updated progress in this area [3].

The stampede of studies in the field of exosomes has flooded valuable information about basic biology and disease into the scientific sphere. We now know that exosomes are more than simple waste receptacles used by cells to rid themselves of unwanted material, but are sophisticated molecular messaging systems that can act locally or distally from where the vesicles are secreted. Exosomal communication is implicated in a myriad biological systems, from immune function and tissue repair [4], nervous system signaling [5], cardiac health [6,7], to more sinister roles in viral pathogeneses such as HIV-1 [8],

Because of the pivotal roles exosomes play in disease, they provide much needed insight into progressing research into avenues such as biomarker identification for diagnostic and prognostic means [9], as well as identifying disease mechanisms as therapeutic targets [10,11]. Cancer is the most studied field where the roles of exosomes have been explored in various processes such as diagnosis, prognosis, metastasis, and therapy [12,13,14,15].

The term asbestos refers to a group of hydrated silica fibers that occur naturally throughout the world. Classified as a category 1 carcinogen [16], asbestos is one of the more notoriously well-known cancer-causing agents. Derived from the Greek word for inextinguishable, asbestos is widely used in the manufacturing process for a multitude of products and thus is prevalent in a significant portion of the world’s communities, particularly in developing nations [17], making it a relevant human health hazard of the present and future [18]. 

Exposure to asbestos occurs overwhelmingly through inhalation and leads to a litany of diseases including lung fibrosis (asbestosis), lung carcinoma, and malignant mesothelioma (MM) [19]. Intriguingly, there is an additive risk of lung cancer when cigarette smoking is combined with asbestos exposure [20]. Asbestos fibers are known to first interact with the upper-respiratory tract and exhibit more lasting effects on lung epithelial cells and resident macrophages, with the fiber geometry dictating how deep into the lungs the asbestos travels (longer, thinner fibers are capable of traveling further) [18,21].

MM is defined as a highly locally-invasive cancer that develops from mesothelial cells that line the body’s cavities. Once exposed to asbestos, there is a remarkably long latency period before MM develops, typically around 10–50 years. Furthermore, once MM is diagnosed, it is fatal within 6–12 months [22]. As noted, the main determinate cause of MM is exposure to asbestos, and unfortunately there are currently no conclusive biomarkers for identifying exposure to asbestos or for the early diagnosis of MM. Moreover, therapeutic strategies for MM are lacking as there are no successful regimens for fighting this disease after its onset, with the chemotherapeutic administration of pemetrexed and cisplatin being the only licensed approach [23]. The mechanism through which this cancer develops in the first place, after asbestos exposure, is also less understood and by delineating the molecular pathways involved, we can gain a foothold of understanding that would no doubt lead to improvements in diagnosis and therapy.

As there are clearly large gaps in the knowledge surrounding MM disease development, onset, treatment, and the minimal presence of potential biomarkers for asbestos exposure and early diagnosis, there exists potential to forward our understanding by delving into the realm of exosome research. This review will provide a brief summary of the current literature and experimental knowledge on MM and asbestos exposure as it pertains to advances in exosome-centered investigations. 

## 2. Malignant Mesothelioma and Exosomes

The first inquiry into exosomes and MM was a focused effort to identify exosomes and their protein cargo from human caner pleural effusions. Exosomes were isolated by sucrose-gradient ultracentrifugation from the pleural fluid of patients suffering from MM, lung cancer, breast cancer, and ovarian cancer. The matrix-assisted laser desorption ionization time-of-flight (MALDI-TOF) mass spectrometric analysis indicated large amounts of peptides originating from immunoglobulins and various complement factors, as well as previously undescribed exosomal proteins such as sorting nexing (SNX25) protein, B-cell translocation gene 1 (BTG1), and pigment epithelium-derived factor (PEDF) [24]. Both BTG1 and PEDF were in increased abundance in exosomes from malignant processes, which may be designated as being involved in tumor exosome biogenesis, according to Bard et al. (2004). Moreover, Western blot analysis verified the presence of the MHC class II molecule, HSP90, and immunoglobulin G and M.

As indicated in the publication, before their results can be generalized, the risk of contaminating proteins that elute with exosomal proteins in these effusions needs to be taken into account. Although pleural effusions contain exosomes from many cellular origins not limited to tumor cells themselves, this report was an important first step in relating exosomes to MM, cancer, and the isolation of possible biomarkers from pleural effusions.

As a follow-up to their first study to entrench upon the paradigm of exosome research in MM, the Lambrecht group [25] conducted a descriptive effort on the protein composition of exosomes that are secreted from MM tumor cells. They chose to study MM due to the limited knowledge of tumor antigens in the disease, and employed MALDI-TOF mass spectrometry to outline the proteomic cargo of MM exosomes. MM tumor cell-lines were created from 10 patients diagnosed with MM, and exosomes were isolated from seven of these tumor cell lines using ultra-centrifugation and were characterized by TEM for their cup-shaped morphology and size range. Exosomal proteins were subjected to MALDI-TOF analysis and of the 38 identified proteins, four were confirmed by Western blot analysis, namely: fascin, β-tubulin, HSC70, and HSP90 [25]. 

In addition, as reported in in vivo systems [26], these tumor exosomes were also enriched with MHC class I molecules, and the authors also indicated high levels of annexins which may be involved in membrane–cytoskeleton dynamics. This report by Hegmans et al. (2004) revealed several proteins that had not yet been indicated on tumor exosomes or in MM cell lines, thus providing novel information on MM and tumor exosomes as a whole that could advance our understanding of the disease.

In 2005, Clayton et al. published their work on the immunological functions of exosomes secreted by tumor cells (breast cancer and mesothelioma), and how these tumor exosomes altered the expression of the NKG2D receptor on target blood leukocytes. The exosomes secreted from these MM cancer cells turned out to be positive in their expression of NKG2D ligands, and this was directly related the capacity of MM exosomes to decrease the capacity of effector T cells to kill target cells [27].

In this study, it was demonstrated that the two MM cell lines used had a high expression of NKG2D ligands (as well as positive staining for MICA, MICB, and ULBP-3), and appeared to correlate with the MM exosomes’ aptitude for more effectively suppressing NKG2D expression on target cells. Overall, this report indicates a role of MM exosomes in phenotypically altering immune cells in a way that can aid tumor cells in immune evasion through the presence of exosome ligands to NKG2D.

A promising field of therapeutic cancer research of late is focused on the use of tumor-associated antigens (TAA) present on tumor exosomes as a mode of dendritic cell-based immunotherapy. The concept being that tumor exosomes bearing TAAs, mostly secreted from immunogenic cancers, are adept at inducing anti-tumor responses in mouse cancer models through the activation of dendritic cells. An intriguing display of this potential was reported by Mahaweni et al. [28], except that by using MM cells, they incorporated a rather unprecedented step forward in this field, because MM is regarded as a non-immunogenic cancer with very few TAAs known. Their investigation assessed if MM exosomes were potential antigen sources for dendritic cell-based immunotherapy [28].

Initially, a lethal dose of MM tumor cells was injected into BALB/c mice. After seven days of tumor formation in the mice, a single bolus dose of dendritic cells was injected into the tumor-bearing mice for immunotherapy. These dendritic cells, however, had been loaded with either MM exosomes or MM cell lysate (or PBS control) to quarry if the exosomes had an immunogenically priming capacity on the dendritic cells. The overall median survival of the tumor bearing mice was significantly increased in the dendritic cell immunotherapy loaded with MM tumor exosomes compared with the cell lysate, indicating that there may be some promise for using MM exosomes as immunotherapy, as well as in other non-immunogenic tumors.

The subsequent research regarding exosomes in MM had an intriguing focus on the formation of tunneling nanotubes (TnTs), the actin-based cellular extensions involved in intercellular cargo transport. The relationship between TnT formation and their communicatory effects with MM tumorigenesis is unknown, and for their study, Thayanithy et al. [29] centered in on exosomes as possible mediators for TnT formation in MM. MM exosomes were purified and added to dishes of independently cultured MM cells, and it was found that in these conditions, MM tumor cells produced significantly more TnTs than the cells cultured without exogenous exosome additions [29]. 

The researchers indicated that the added tumor exosomes were enriched at the base of, and inside the TnTs, which was correlated interestingly to a 2016 report [30] on the mode of exosomal interaction with target cells. In the study by Heusermann et al., exosomes were demonstrated to localize and “surf” on the filipodia (similar actin filamentous cellular projections) before internalization [30,31]. The uptake of MM exosomes by MM cells apparently facilitated more TnT connections between the tumor cells, and the connected cells had nearly twice the number of lipid-raft enriched regions. Taken together, it can be seen that MM exosomes may act as an induction agent of TnT formation between MM tumor cells, and perhaps this connection is an important conduit of cellular information vital to MM progression.

Progressing on the understanding of the MM secretome, Greening et al. [32] released a comprehensive study on MM-derived exosomal proteomic cargo. Through the use of quantitative proteomics, they delineated the protein make up of exosomes from four human MM cell lines and identified a total of 2178 proteins from all cells, with 631 common exosomal proteins between all groups [32]. As this report came after the aforementioned exosomal inquiries in MM, there were several common proteins identified in the previous report [24]; however, 2073 proteins were unique to this investigation. Of their MM exosome proteins, the investigators demarcated candidate biomarkers based on clinical relevance, amongst them being tubulin isotypes TUBB4A, Q8IWP6, and B3KPS3; galectin-3-binding protein and LGB3P; and alpha enolase, annexin 1, and G6PD. Furthermore, it was identified that MM exosomes contained mesothelin, calreticulin, vimentin, and superoxide dismutase, all known to be highly expressed in MM. Additionally, the results of this research uncovered the presence of 26 immunoregulatory components in MM exosomes (such as oncostatin-M receptor (OSMR), multidrug resistance-associated protein 1 (ABCC1), and the SUMO-1 activating receptor (SAE1)), as well as 16 tumor-derived antigens, including glypican-1, which has been identified in many tumor-derived exosomes and has been recorded as a potentially valuable biomarker for pancreatic cancer [33]. Importantly, this study also provided valuable insight that showed that MM exosomes regulate the cells of the tumor microenvironment by increasing the migratory capacity of fibroblasts and endothelial cells in vitro. Together, their findings implicate MM exosomes as containing many proteins relevant to cancer, angiogenesis, metastasis, migration, and immune regulation.

The Robinson group provided another iteration on their quests for elucidating the complexities of the MM secretome using iTRAQ proteomic analysis. Using 6 MM cell lines in comparison with three primary mesothelial cell cultures, it was seen that MM cell secretomes contained higher abundances of exosomal proteins [34]. This study is primarily focused on the whole secretome, with only some references to the exosomes. A study by Javadi et al. utilizing a small number of patient samples demonstrated the potential utility of extracellular vesicles (including, exosomes, microvesicles, and apoptotic bodies) in diagnosing benign vs. malignant MPM [35]. The ratios of mesothelin, galectin-1, osteopontin, and VEGF were higher in MPM samples compared with the benign effusion, whereas exosomal angiopoietin-1 was higher in the benign samples compared with the malignant ones. The findings are encouraging and need to be validated with larger sample populations.

Although more emphasis has been put on the exosomal proteomic signature, a report by Cavalleri et al. (2017) [36] suggests that a specific exosomal microRNA signature can discriminate malignant pleural mesothelioma (MPM) from past asbestos exposure (PAE) subjects. This study was done in small number of subjects and needs to be verified in larger cohorts. Later, a group using the MM tumor stromal model demonstrated that endothelial cell-derived exosomes enriched in miR-126 were differentially distributed within the stroma. The findings of the study communicated an important role regarding the exosomal transfer of miR-126 in its anti-tumor response in MM [37]. The same group further demonstrated that MPM-derived spheroids, when treated with the miR-126-enriched exosome showed an anti-tumor effect initially. However, later, the effect was vanished due to loss of miR-126 from cells that could be restored by inhibition of exosome secretion [38].

The literature review presented above is 100% focused on exosomal content/signature from MM cells and how exosomes can help in communication between MM cells. However, the role exosomes can play in the development of MM or in assisting in the early diagnosis of MM is limited. For more than a decade, our lab has been interested in uncovering the mechanisms of MM development in response to asbestos exposure. Based on the fact that asbestos is inhaled into the lungs, yet MM develops in remotely present pleural and peritoneal mesothelial cells, we were encouraged to focus on exosomes as a carrier of information from lung cells to mesothelial cells. As a first of its kind study, our lab investigated the proteomic cargo and gene modulatory effects of exosomes from asbestos-exposed cells. Our investigation began by culturing lung epithelial cells (BEAS2B) or macrophages (THP1) (the first known cells to encounter asbestos upon inhalation) with asbestos and isolating their exosomes. These asbestos-exosomes were subjected to tandem-mass spectrometry for protein identification. It was shown that 145 proteins were identified in epithelial cell exosomes and 55 were significantly different in abundance in the asbestos-exposed group, including plasminogen activator inhibitor 1, vimentin, thrombospondin, and glypican-1 [39]. We next found that the exosomes from asbestos-exposed epithelial cells led to genetic changes in the target primary pleural human mesothelial cells (HPM3), which were reminiscent of epithelial to mesenchymal transition (EMT): down-regulation of E-cadherin, desmoplakin, and the IL1 receptor antagonist [39]. 

Upon proteomic analysis of the macrophage exosomes, we (Munson et al., 2018) identified 785 proteins. Of these proteins, 32 had significantly different abundances between the exosomes from the asbestos-exposed group and the control. Fifteen of these exosomal proteins were in greater abundance in the asbestos group compared with the control, and interestingly, vimentin and SOD were among those that showed an increase in exosomes from the macrophages after asbestos exposure. In response to exposure of asbestos exosomes from macrophages to target primary mesothelial cells, it was shown that significant genetic alterations occurred in the mesothelial cells: 498 gene changes in total (with 1.5-fold cutoff with an ANOVA transcript level *p*-value less than 0.05), with 241 up and 257 down-regulated. As a positive control, the group used asbestos fibers on mesothelial cells, and uncovered that 206 genes were mutually altered in the asbestos-exposed exosomes or asbestos-exposed group of mesothelial cells. Three up- (*hCCNB2*, *hEGR1* and *hFANCD2*) and down-regulated (*hCRELD2*, *hERO1B* and *hJAG1*) genes were then validated by qPCR [39]. Of note is that CCNB2 overexpression was attributed to MM and FANCD2 as up-regulated during MM and asbestos exposure [40,41]. This exciting discovery is novel in that it implicates exosomes from asbestos-exposed cells as being capable of changing mesothelial cell genetics in ways similar to how asbestos fibers would change on their own. As a next step, this information will be verified in in vivo systems for future studies.

As an initial step in the direction of in vivo study, we committed our efforts to defining the proteomic signature of mouse serum exosomes in an asbestos-exposure model. C57/Bl6 mice were exposed to asbestos via oropharyngeal aspiration, and 56 days later, the serum exosomes were isolated for proteomic analysis. Again, using tandem-mass spectrometry for protein identification, we showed that there were 376 quantifiable proteins in the mouse serum exosomes, with the majority of protein being more abundant in the asbestos-exposed group [42]. Of these more abundant proteins in the asbestos-exposed group, three were validated by Western blot analysis, all of which were acute-phase proteins: haptoglobin; ceruloplasmin, the copper carrying glycoprotein previously seen to be increased in MM patients’ blood and asbestos exposed individuals [43]; and fibulin-1, a member of the fibulin family, of which, fibulin-3 has been suggested as being implicative of asbestos exposure and MM [44]. We did not see common exosomal proteins between our two published studies as these were very different systems, namely in vitro vs. in vivo and human vs. mouse.

Our 2019 [45] findings on the secreted exosomes of mesothelioma cells compared with healthy mesothelial cells showed that the tumor cells secreted significantly different patterns of miRNAs compared with their healthy counterparts. In particular, we pointed out that miR-16-5p was significantly increased in expression within the exosomes released by the cancer cells. Our hypothesis was that the mesothelioma cells developed a preferential secretion mechanism to rid themselves of miR-16-5p due to its well-established tumor suppressor functions. Multiple experiments indicated the functionality of this secretion and the possibility of targeting this pro-tumor phenotype. Additionally, we are one of the few groups to posit this mode of tumor progression through the active release of tumor suppressors such as miR-16-5p.

Furthermore, our 2020 study [46], where multiple cancer researchers considering a number of different human cancers performed a comprehensive proteomic analysis of the extracellular vesicles and particles (EVPs), demonstrated that EVP proteins can be used for cancer-type detection. Focusing on the mesothelioma data of the paper showed that immunoglobulins were the top family of proteins found in EVP at a high frequency in th mesothelioma. The study suggested that plasma-derived EVP protein signatures could be beneficial for cancer-type detection in patients. These findings, although encouraging, need further validation and testing in a lager cohort of patients to confirm the results.

The studies discussed above are summarized in a table (Table 1). A schematic (Figure 1) based on our findings is included to explain how exosomes can help in the development and survival of MM.

In addition to the above-mentioned published studies, we also performed numerous studies with human mesothelioma cells, plasma from asbestos-exposed samples, and mesothelioma patient samples. We did find some common signatures between our study and others, including SOD, vimentin, and glypican-1 [32,34]. Studies were also performed with plasma exosomes isolated from healthy volunteers, the asbestos-exposed non-tumor group, and the asbestos-exposed mesothelioma groups. Although the number of exosomes per ml of plasma were not different in various groups, the exosomal protein quantity was more in different disease groups compared with the controls. The proteomic analysis performed on these samples showed the presence of coagulation-related proteins in the exosomes from the disease group (mesothelioma and asbestos-exposed) compared with the control. The control group plasma exosomes presented a signature including immunoglobulins, lipoproteins, and platelet-related proteins. These data indicate altered immune surveillance in MM samples concomitant with the increase of coagulation factors (unpublished data). We planned to repeat/validate these studies with a larger sample size before publishing the data.

## 3. Conclusions

Asbestos exposure is a serious health concern for thousands of people worldwide, and MM is the cancer resulting primarily from asbestos exposure. To date, there are no successful therapeutic regimens for treating MM, and the dismal survival time after diagnosis and lack of biomarkers for early detection make it an important area for propagating research. The field of exosomes in cancer has exploded recently due to the fact that these extracellular vesicles are emerging players in the dynamics of cancer biology, contributing to the cellular crosstalk involved in cancerous processes and housing cancer biomarker signatures. Many advances have been made to date using exosomes for biomarker identification, detecting novel therapeutic targets, and basic understanding of tumor biology [47,48]. In this regard, using exosomes to gain the needed insight into MM development, detecting potential biomarkers, pinpointing therapeutic targets, and harnessing exosomes as drug delivery devices and immune-regulators against cancer are the next big steps researchers need to take.

The studies reviewed above provide the initial framework for understanding possible biomarkers and the underlying biology of MM and asbestos exposure. From their findings, research can attempt to further identify the means of early detection of asbestos exposure or asbestos-related disease development, as well as uncover the much needed therapeutic targets. Moreover, the ability to understand the mechanistic of MM development and progression regarding exosomes is an important realm that may be utilized for treating MM cancer patients. Ultimately, we hope that exosome research in MM will continue on this forward trajectory and that more significant findings will be made for figuring out how asbestos causes cancer, and finding ways to identify dangerous exposure to asbestos and early cancer detection before a fatal diagnosis is made.

Although highly prolific, the field of exosome research has offered many opportunities for medical field advancement, but it is not without challenges and limitations. Some areas of improvement include methods of exosome isolation, understanding the mechanisms of biogenesis, and the characterization of exosome cargo. The advances being made in the field are remarkable and give us hope for potential breakthrough treatments and/or technologies.

## Figures and Tables

**Figure 1 ijms-23-15438-f001:**
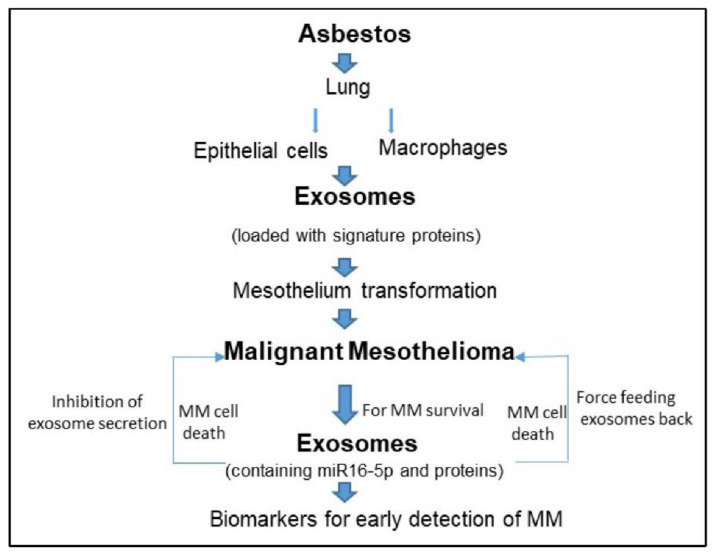
Asbestos inhalation results in lung epithelial cells and macrophages to release exosomes, which are loaded with a special protein and RNA molecules. These exosomes can travel to the mesothelium and transform them so as to develop malignant mesothelioma. Malignant mesothelioma cells/tumors secrete exosomes that are loaded with miR16-5p (tumor suppressor miRNA) to survive. Inhibition of exosome secretion or feeding them back to MM cells results in MM cell death. Protein signature in secreted MM tumor exosomes can be utilized to detect MM.

**Table 1 ijms-23-15438-t001:** A summary of exosome-related studies in the field of malignant mesothelioma (MM).

MM Samples/Specimens	Molecules (RNA, Proteins) in Exosomes	Core Findings	Reference
Human pleural effusions	Proteins and peptides	Known and new exosomal proteins SNX25, BTG1, PEDF, identified as biomarkers.	Bard et al., 2004 [24]
Seven humani MM tumor cell lines	Proteomic cargo	Identified 38 proteins, confirmed 4 by immunoblot. New information on MM exosomes	Hegmans et al., 2004 [25]
MM cell lines	Proteins, ligands	Immunological function of exosomes via NKG2D ligand expression	Clayton et al., 2005 [27]
MM tumor bearing mice	Tumor associated antigens (TAA)	MM exosome show some promise in immunotherapy	Mahaweni et al., 2013 [28]
MM cell lines and MM exosomes	Tunneling nanotubes (TnTs)	MM tumor cells form more TnTs in presence of exogenous exosomes. TnTs help in intercellular cargo transport	Thayanithy et al., 2014 [29]
Four human MM cell lines	Proteins	Identified 2073 unique proteins in MM exosomes. Proteins could play important roles in cancer angiogenesis, metastasis, migration and immune regulation	Greening et al., 2016 [32]
Six MM cell lines and 3 primary mesothelial cell lines	MM secretome proteins	MM secretome contained high abundance of exosomal proteins	Creaney et al., 2017 [34]
Small cohort	microRNA	Specific exosomal miRNA signature can discriminate MM from past asbestos exposure subjects	Cavalleri et al., 2017 [36]
Asbestos exposed lung epithelial cells, macrophages and mesothelial cells	Proteins and genes	Exosomes from asbestos exposed lung epithelial cells and macrophages can cause gene expression changes in mesothelial cells.	Munson et al., 2018 [39,42]
Blood/serum from asbestos exposed mice	Proteins	Asbestos exposure caused increased abundance of certain proteins in serum exosomes.	Munson et al., 2019 [45]
Human mesothelioma and mesothelial cells	microRNA	MM cells preferentially secrete tumor suppressor miRNA, miR16-5p via exosomes. Force feeding exosomes or inhibition of secretion of exosome is beneficial for death of MM tumor cells.	Munson et al., 2019 [45]
Plasma from human MM patients	proteins	Exosome isolated were enriched with immunoglobulins. Study demonstrated that plasma-derived exosomal protein signature could be beneficial for cancer type detection	Hoshino et al., 2020 [46]
Small number of MM patient sample	Proteins	Utility of exosomal protein signature in diagnosis of benign vs. malignant MM.	Javadi et al., 2021 [35]
MM tumor stromal model. MM spheroids	microRNA	Exosomal transfer of miR-126 plays an anti-tumor response in MM.	Monaco et al., 2019 [37] Monaco et al., 2022 [38]

## Data Availability

Not applicable.

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
