# Peer review of "Potential Roles of Exosomes in the Development and Detection of Malignant Mesothelioma: An Update"

_ijms, 2022, doi:10.3390/ijms232315438_

Round 1

Reviewer 1 Report

The article by Shukla et al., is well-organized and a novel piece of work. The author needs to add the following to make the article ready for publication:

1. Basic introduction for extracellular vesicles mentioning size based classification and origin.

2. Different methods used to detect EVs.

3. A basic flow diagram of discovery and usage of EVs in asbestos and a flowdiagram for asbestos detection to summarise the study.

Author Response

  1. Basic introduction for extracellular vesicles mentioning size based classification and origin. Added in line 24-29.

2. Different methods used to detect EVs. Added lines 30-32 with a reference.

3. A basic flow diagram of discovery and usage of EVs in asbestos and a flowdiagram for asbestos detection to summarise the study. Added as a figure.

Reviewer 2 Report

Munson and Shukla present a thorough review of the current knowledge elucidating the possible roles of exosomes in the early detection, development and therapy of malignant mesothelioma (MM). The manuscript is well-written and each reviewed study was described meticulously.

Specific comments

Title. The current title is too general and does not acknowledge the impact of the manuscript relative to its relevance in early detection, disease development and clinical management of MM as stated by the authors under Conclusion (line 282-297). Kindly revise the title as appropriate.  The authors may wish to change “mesothelioma” to malignant mesothelioma as has been used in the entire manuscript.

Abstract. The authors should include strong reasons how this review will advance the field of MM management relative to their statements in line 8-13.

To provide a better overview of the reviewed studies showing the different potential roles of exosomes in MM, it is essential to summarize the core findings, including the specific molecules/miRNAs, experimental approach (in vitro or in vivo) and specimens implicated in each study in a summary table.

Conclusion. Based on reviewed literatures, the authors made strong statements backing up the potentials of exosomes in many biological/clinical aspects of MM, hence, indicating their relevance in the management of this malignancy. Are there no shortcomings or challenges to consider in this dominant view to build on this topic in the future?

Author Response

Title. The current title is too general and does not acknowledge the impact of the manuscript relative to its relevance in early detection, disease development and clinical management of MM as stated by the authors under Conclusion (line 282-297). Kindly revise the title as appropriate.  The authors may wish to change “mesothelioma” to malignant mesothelioma as has been used in the entire manuscript. The title has been changed as per suggestion of the reviewer.

Abstract. The authors should include strong reasons how this review will advance the field of MM management relative to their statements in line 8-13. Lines 15-19 have been included as per reviewer's suggestion, stating how our findings can potentially advance the field of MM management.

To provide a better overview of the reviewed studies showing the different potential roles of exosomes in MM, it is essential to summarize the core findings, including the specific molecules/miRNAs, experimental approach (in vitro or in vivo) and specimens implicated in each study in a summary table. A summary table has been included.

Conclusion. Based on reviewed literatures, the authors made strong statements backing up the potentials of exosomes in many biological/clinical aspects of MM, hence, indicating their relevance in the management of this malignancy. Are there no shortcomings or challenges to consider in this dominant view to build on this topic in the future? Shortcomings and challenges of the field has been included at the end in lines 312-316.